# Anti-inflammatory effect of the combined treatment of LMT-28 and kaempferol in a collagen-induced arthritis mouse model

Young-Jin Jeong[1]☯, Sun-Ae Park[1]☯, Yeon-Hwa Park[2], Lee Kyung Kim[1], Hae-Ri Lee[1], Hee Jung Kim[1]*, Tae-Hwe Heo◯[1]*

1 Laboratory of Pharmacoimmunology, Integrated Research Institute of Pharmaceutical Sciences and BK21 FOUR Team for Advanced Program for Smart Pharma Leaders, College of Pharmacy, The Catholic University of Korea, Bucheon-si, Gyeonggi-do, Republic of Korea, 2 Biowave, Anyangcheon-ro, Yangcheon-gu, Seoul, Republic of Korea

☯ These authors contributed equally to this work.
* thhur92@catholic.ac.kr (THH); hjk0114@catholic.ac.kr (HJK)

**Data Availability Statement:** All relevant data are within the paper and its Supporting information files.

## Abstract

Rheumatoid arthritis (RA) is an autoimmune disease characterized by joint inflammation and swelling. Several studies have demonstrated that RA fibroblast-like synovial cells (RA-FLS) play an important role in RA pathogenesis. Activated RA-FLS contribute to synovial inflammation by secreting inflammatory cytokines including interleukin (IL)-1β, IL-6 and tumor necrosis factor-α. LMT-28 is derivative of oxazolidone and exerts anti-inflammatory effects on RA via IL-6 signaling pathway regulation. LMT-28 also regulates T cell differentiation in RA condition. However, the effect of LMT-28 on the migration and invasion of RA-FLS remains unknown. Kaempferol has been reported to have pharmacological effects on various diseases, such as inflammatory diseases, autoimmune diseases, and cancer. Additionally, kaempferol has been reported to inhibit RA-FLS migration and invasion, but it is not known about the therapeutic mechanism including molecular mechanism such as receptor. The present study aimed to investigate the synergistic effects of the combined treatment of LMT-28 and kaempferol on RA-FLS activation and RA pathogenesis in mouse model. LMT-28 and kaempferol co-administration inhibited RA disease severity and histological collapse in the joint tissues of CIA mice, as well as downregulated the levels of pro-inflammatory cytokines in mouse serum. Additionally, the combined treatment inhibited excessive differentiation of T helper 17 cells and osteoclasts. Furthermore, compared with single treatments, combined treatment showed enhanced inhibitory effects on the hyperactivation of IL-6-induced signaling pathway in RA-FLS. Combined treatment also inhibited RA-FLS cell proliferation, migration, and invasion and suppressed the expression of matrix metalloproteinase in RA-FLS. Furthermore, we confirmed that the combined treatment inhibited chondrocyte proliferation, migration, and invasion. In conclusion, our results suggest that the combined treatment of LMT-28 and kaempferol exerts a synergistic effect on the RA development via the regulation of IL-6-induced hyperactivation of RA-FLS. Furthermore, this study suggests that combination therapies can be an effective therapeutic option for arthritis.

**Funding:** This work was supported by the Basic Science Research Program through the National Research Foundation of Korea (NRF) funded by the Ministry of Education, Science, and Technology [grant number. 2018R1A6A1A03025108 and 2022R1A2C2009911]. The funders had no role in study design, data collection and analysis, decision to publish, or preparation of the manuscript.

**Competing interests:** The authors have declared that no competing interests exist.

## Introduction

Rheumatoid arthritis (RA) is a chronic autoimmune inflammatory disease accompanied by synovial tissue destruction and dysfunction [1]. The development of arthritis is induced by various factors including genetics, environment, and lifestyle [2]. The common clinical symptoms of RA include extra-articular synovitis, joint and tendon inflammation, malaise, fatigue, and fever [3]. Synovitis, characterized by immune cells infiltration into the synovial membrane, is a hallmark of RA [4]. Pathogenic T lymphocytes cause synovial inflammation by secreting pro-inflammatory cytokines [5]. A subset of CD4$^+$ effector cells, termed T helper 17 (Th17) cells, secrete interleukin (IL)-6 and IL-17A in inflamed tissue [6]. IL-6 and transforming growth factor (TGF)-β induce the expression of retinoid-related orphan receptor gamma T (ROR-γt), a key transcription factor for Th17 cell differentiation [7]. Additionally, IL-6 upregulates the expression of receptor activator of nuclear factor-kappa B ligand (RANKL) in synovium [8]. The binding of RANKL and RANK activates the extracellular signal-regulated kinase (ERK), AKT, and p38 mitogen-activated protein kinase (MAPK) signaling pathways [9] and enhances the expression of osteoclast-related transcription factors c-Fos and nuclear factor of activated T cells (NFATc1) [10]. RANKL-induced differentiation of osteoclast precursor cells leads to bone resorption and degeneration in joint tissue [11].

The activation of RA-fibroblast-like synovial cells (RA-FLS) plays an important role in RA pathogenesis and synovial inflammation [12]. Activated RA-FLS induces the secretion of matrix metalloproteinases (MMPs) that degrade the extracellular matrix [13]. Overproduction of MMPs causes RA-FLS to migrate and invade joint tissue and destroy the cartilage and bone in patients with RA [14]. Previous studies have shown that MMP levels are increased in the synovial tissue of patients with RA [15], and MMP-3 and MMP-9 significantly contribute to RA-FLS activation [16]. Activated RA-FLS also secrete immunomodulatory factors, including inflammatory cytokines, which participate in RA pathogenesis by inducing and maintaining synovial inflammation [17].

Inflammatory cytokines are key players in the initiation and progression of RA [18]. The levels of IL-1β, IL-6, and tumor necrosis factor (TNF)-α elevated in the serum and synovial fluid of patients with RA [19]. IL-6 is a representative pro-inflammatory cytokine that induces RA-FLS activation and causes chronic inflammation in joint tissues [20]. Inflammatory and degenerative arthritis is characterized by excessive activation of IL-6 trans-signaling [21]. Additionally, the concentrations of IL-6 and soluble IL-6 receptor (sIL-6R) are increased in RA patients, which is associated with disease severity in RA [22]. The IL-6/IL-6R complex induces glycoprotein 130 (gp130) dimerization, promoting the activation of the downstream signaling pathways Janus kinase (JAK) and signal transducer and activator of transcription (STAT) [23]. IL-6/IL-6R/gp130 complex-induced activation of the JAK/STAT signaling pathway plays an important role in cell differentiation, proliferation, and apoptosis and regulates various immune functions during RA [24]. STAT signaling pathway plays a critical role in the differentiation of CD4$^+$ effector T cells and the expression of the ROR-γt transcriptional factor of Th17 cells [25]. Additionally, the phosphorylation of STAT3 inhibits RA-FLS apoptosis and induces the proliferation, survival, migration, and invasion of RA-FLS [26]. IL-6-induced activation of MAPK signaling pathway is also involved in RA-FLS mobility and MMP expression [27]. Based on the biological activities of IL-6 in RA development, therapies that directly target IL-6 or signaling pathways are widely used as treatment strategies for RA [28, 29]. A previous study demonstrated that the inhibition of the ERK and p38 MAPK signaling pathways can suppress the migration and invasion of RA-FLS [30].

Kaempferol, used as a combination partner of LMT-28 in this study, is a type of flavonoid and is reported to have anti-inflammatory and anticancer effects in various immune diseases

[31]. Additionally, kaempferol inhibited the migration and invasion activity of RA-FLS by inhibiting the activation of the MAPK pathway [32]. In our previous study, we demonstrated that LMT-28 exerts a therapeutic effect on RA development by inhibiting the IL-6-induced signaling pathways [33]; however, it remains unclear whether LMT-28 has a regulatory effect on RA-FLS migration and invasion. Based on previous research, this study designed the combined administration of LMT-28 and kaempferol to confirm further improved preventive effects of combination on the hyperactivation of RA-FLS and RA development by simultaneously targeting the JAK/STAT pathway and MAPK pathway, which are major signaling pathways that play an important role in arthritis development. Our results demonstrated that combining LMT-28 with kaempferol can effectively alleviate arthritis symptoms in mice by downregulating the RA-FLS activation.

## Materials and methods

### Cell culture

The human RA-FLS cell line, MH7A, was provided by Prof. Yong Yeon Cho (The Catholic University of Korea, Gyeonggi-do, Republic of Korea). Human chondrocyte cell line, C28/I2 cell, was purchased from American Type Culture Collection (ATCC, Manassas, VA, USA). MH7A cells were cultured in RPMI 1640 medium (Welgene, Gyeongsangbuk-do, Republic of Korea) and C28/I2 cells were cultured Dulbecco's Modified Eagle's Medium (Corning, NY, USA), respectively, supplemented with 10% fetal bovine serum (Corning) and 1% penicillin/streptomycin (Thermo Fisher Scientific, Waltham, MA, USA), and incubated at 37°C in 5% humidified $CO_2$ incubator.

### Animals

Six-week-old male DBA/1J and C57BL/6 mice were purchased from Orient Bio (Gyeonggi-do, South Korea). The animal experiments were conducted in accordance with internationally accepted standards and protocols and were approved by the Institutional Animal Care and Use Committee of the Catholic University of Korea, Songsim campus (Bucheon-si, Gyeonggi-do, Korea) (approval number: 2016-018-02).

There is no experimental method in this experiment that causes extreme stress or pain, but joint pain and discomfort may occur due to arthritis development. Therefore, the condition of the experimental animals was monitored in real time, and to alleviate the pain of the experimental animals, euthanasia was performed when the mouse arthritis score first reached 4 points. At the end-time, mice were placed in euthanasia chamber and filled with CO2 at a rate of 10–30% per minute to euthanasia and checked for loss of consciousness, breathing, and cardiac arrest in the mice after euthanasia.

### Collagen-induced arthritis model and evaluation of arthritis score

To evaluate the synergistic effects of LMT-28 and kaempferol on RA-FLS activation and RA pathogenesis, a collagen-induced arthritis (CIA) mouse model was established as described below. To provide the first immunization, 50 μg bovine type II collagen (Chondrex, Woodinville, WA, USA) and complete Freund's adjuvant (Sigma-Aldrich, St. Louis, MO, USA) were intradermally injected in DBA/1J mice through the base of the tail. After 2 weeks, mice were immunized with 50 μg bovine type II collagen and incomplete Freund's adjuvant (Sigma-Aldrich). Following the second immunization, mice received oral administration of LMT-28 (1 mg/kg) (Sigma-Aldrich), kaempferol (10 mg/kg) (Sigma-Aldrich), a combination of LMT-28 (1 mg/kg) and kaempferol (10 mg/kg), or vehicle (10% DMSO in distilled water) (N = 10 in

**Table 1. CIA scoring system.**

| Score | Typical mouse paw appearance |
|---|---|
| 0 | Normal paw |
| 1 | One or two toes inflamed and swollen |
| 2 | More than three toes inflamed w/no paw swelling, or mild swelling of entire paw |
| 3 | Entire paw inflamed and swollen |
| 4 | Severely swollen paw and all toes, or ankylosed paw and toes |

each group) six times per week. The severity of arthritis in the fore and hind paws was determined twice a week from day 0 to 63 after the first immunization. The arthritis score was evaluated by three blinded observers according to the Hooke Laboratories Manual, as described in Table 1.

## Joint tissue staining

The hind paws were dissected and decalcified in 10% formic acid (Sigma-Aldrich) for 21 days. After decalcification, specimens were dehydrated using ethanol and xylene (DUKSAN, Gyeonggi-do, South Korea) and embedded in paraffin pastilles (Merck Millipore) overnight. Paraffin blocks were sectioned into 4 μm slices, which were then dried on a glass slide overnight. The tissue sections were deparaffinized and rehydrated with xylene and ethanol. After staining with EASY STAIN Harris hematoxylin (YD Diagnostics, Gyeonggi-do, South Korea) and Eosin Y solution (MUTO PURE Chemicals, Tokyo, Japan), the tissue sections were dehydrated with ethanol and xylene again. The sections were mounted onto the slides using a mounting solution (Thermo Fisher Scientific) and imaged using an APERIO CS2 slide scanner (Leica, Wetzlar, Germany). Histological analysis of joint tissues stained with hematoxylin and eosin was performed by Orient GENIA (Gyeonggi-do, Republic of Korea) according to manufacturer's guidelines and described in Table 2.

**Table 2. Histopathological analysis of joint tissues of CIA mice.**

| Score | Histological feature | Description |
|---|---|---|
| 0 | Inflammation | Normal |
| 1 | | Slight inflammation |
| 2 | | Moderate inflammation |
| 3 | | Severe inflammation |
| 0 | Synovial hyperplasia | Normal |
| 1 | | Slight synovial hyperplasia |
| 2 | | Moderate synovial hyperplasia |
| 3 | | Severe synovial hyperplasia |
| 0 | Pannus formation | Normal |
| 1 | | Slight pannus formation |
| 2 | | Moderate pannus formation |
| 3 | | Severe pannus formation |
| 0 | Erosion of cartilage and bone | Normal |
| 1 | | Slight erosion |
| 2 | | Moderate erosion |
| 3 | | Severe erosion |

## Enzyme-linked immunosorbent assay (ELISA)

Antibodies against TNF-$\alpha$, IL-1$\beta$, IL-6, and IL-17A (Biolegend, San Diego, CA, USA) were diluted in 1× phosphate-buffered saline (PBS) and added to Immuno 96-well plates (Thermo Fisher Scientific) at 100 μL/well. The plates were incubated at 4°C overnight and treated with 200 μL/well of blocking buffer (1% BSA dissolved in 1× PBS) for 1 h. Subsequently, 100 μL of diluted serum or soup of MH7A culture medium and standard (0–500 pg/mL) were added to the wells. Avidin-horseradish peroxidase-conjugated secondary antibody (BioLegend) was added for 30 min, and TMB substrate solution (Surmodics, Eden Prairie, MN, USA) was added to each well until sufficient color development was confirmed. After stopping the reaction with 2N HCl, a microplate reader (BioTek, Winooski, VT, USA) was used to measure the optical density at 450 nm.

## *Ex vivo* mouse Th17 cell differentiation assay

Spleens isolated from normal DBA1/J mice were separated into single cells. Splenocytes were then incubated with 1× RBC lysis buffer (eBioscience, San Diego, CA, USA) at room temperature for 5 min. After washing with PBS, naïve CD4+ T lymphocytes were isolated using mouse CD4 (L3T4) MicroBeads (Miltenyi Biotec, Bergisch Gladbach, Germany). A Th17 cell differentiation assay with mouse splenocytes was performed using a mouse Th17 differentiation kit (R&D Systems, Minneapolis, MN, USA) according to the manufacturer's instructions. Briefly, hamster anti-mouse CD3 antibody (BioLegend) and anti-mouse CD28 antibody (BioLegend) were added to 96-well tissue culture plates, which were incubated at 4°C overnight. The naïve CD4+ T lymphocytes isolated from the spleen were suspended in X-VIVO™ 15 medium added at 125 μL/well (Lonza, Switzerland), seeded into anti-mouse CD3 and CD28 antibody-coated wells, and incubated at 37°C with 5% $CO_2$ overnight. Stimulated CD4+ T lymphocytes were treated with LMT-28 (25 μM) and/or kaempferol (12.5 μM), and the plates were incubated at 37°C in 5% $CO_2$ for 3 days. All the diluted reagents in the medium were refreshed every 2 days, and 5 days after stimulation, cells were collected and washed with X-VIVO™ 15 medium. Cells were suspended with X-VIVO™ 15 media containing 1× cell stimulation cocktail (Thermo Fisher Scientific) and 1× monensin (eBioscience) for stimulation and incubated at 37°C in 5% $CO_2$ for 4 h. Subsequently, the cells were analyzed using flow cytometry.

## Flow cytometry analysis

Differentiated CD4+ T cells and splenocytes isolated CIA mice were centrifuged and incubated with anti-CD4-PE-cy7 antibody (Thermo Fisher Scientific) or anti-CD25-VioBright FITC antibody (Miltenyi Biotec) for 30 min at 4°C in the dark. After washing, cells were fixed/permeabilized using the Intracelluar Fixation & Permeabilization buffer set (eBioscience) at 4°C for 10 min in the dark. The cells were incubated with anti-IL-17A-PE antibody (Miltenyi Biotec) or anti-mouse FOXP3 antibody (Invitrogen, Waltham, MA, USA) diluted in 1× permeabilization buffer (eBioscience) for 30 min at 4°C in the dark. After washing, the cells were resuspended in FACS buffer and analyzed using a flow cytometer (FACS CANTO II; BD Biosciences, Franklin Lakes, NJ, USA).

## Osteoclastogenesis assay

Mouse bone marrow-derived macrophages (BMMs) isolated from normal C57BL/6 mice were seeded on cell culture plates in α-MEM media (Thermo Fisher Scientific) containing 20 ng/mL of macrophage colony-stimulating factor (M-CSF; R&D Systems) and incubated at 37°C in 5% $CO_2$ for 4 days. After incubation, BMMs were treated with M-CSF (20 ng/mL), RANKL

**Table 3. RT-qPCR primers used in this study.**

| Gene | Primer sequence (5′-3′) | | Amplicon length (bp) |
|------|-------------------------|--|----------------------|
|      | Forward | Reverse | |
| c-fos | GCGAGCAACTGAGAAGAC | TTGAAACCCGAGAACATC | 162 |
| nfatc1 | CAACGCCCTGACCACCGATAG | GGCTGCCTTCCGTCTCATAGT | 395 |
| ctsk | ATACGTTACTCCAGTCAAGAACCAG | ATAATTCTCAGTCACACAGTCCACA | 151 |
| trap | GGCTACTTGCGGTTTCACTATG | GGGAGGCTGGTCTTAAAGAGTG | 131 |
| Mmp-1 | CTGTTCAGGGACAGAATGTGCT | TCGATATGCTTCACAGTTCTAGGG | 85 |
| Mmp-2 | CCGTCGCCCATCATCAAGTT | CTGTCTGGGGCAGTCCAAAG | 169 |
| Mmp-3 | TATGGACCTCCCCCTGACTCC | CAGGTTCAAGCTTCCTGAGG | 187 |
| Mmp-9 | CGTGTCTGGAGATTCGACTTGA | TTGGAAACTCACACGCCAGA | 165 |
| Mmp-13 | CTGGCCTGCTGGCTCATGCTT | GCAGGGTCCTTGGAGTGGTCA | 166 |
| Gapdh | CGAGATCCCTCCAAAATCAA | TTCACACCCATGACGAACAT | 170 |

(40 ng/mL; R&D Systems), LMT-28 (25 μM) and/or kaempferol (12.5 μM) for 3 days. Media were replaced with α-MEM media containing same reagents and incubated for another 2 days. The cells were stained for analyze the tartrate-resistant acid phosphate (TRAP) activity using a TRAP single-stain kit (TAKARA Bio, Shiga, Japan) according to the manufacturer's instructions. TRAP+ cells with three or more nuclei were considered mature osteoclasts.

## Quantitative reverse transcription polymerase chain reaction

Total RNA was extracted from MH7A and C28/I2 cells using TRIzol reagent. cDNA was synthesized from RNA (1 μg) using the PrimeScript™ RT Master Mix cDNA synthesis kit (TAKARA Bio). Quantitative reverse transcription polymerase chain reaction (RT-qPCR) was performed using TB Green Premix Ex Taq II (TAKARA Bio), according to the manufacturer's protocol. Briefly, initial denaturation was performed at 95˚C for 30 s, followed by 40 cycles of thermal cycling at 95˚C for 5 s and 60˚C for 30 s. The relative mRNA levels were normalized to those of GAPDH. The primers used for RT-qPCR are listed in Table 3.

## Western blot assay

The total lysates of MH7A cells and differentiated osteoclasts were used for western blotting. Cells were pre-treated with LMT-28 (25 μM), kaempferol (12.5 μM), or combination (LMT-28 25 μM + kaempferol 12.5 μM) for 1 h and then stimulated with 20 ng/mL hyper IL-6 (Recombinant human IL-6/IL-6R alpha protein chimera; R&D Systems) for 5 min. Cells were lysed using radioimmunoprecipitation assay lysis buffer (Biosesang, Gyeonggi-do, South Korea). Proteins extracts were loaded in and separated using SDS-PAGE gel and transferred to a MembraneImmobilon®-P transfer membrane (Merck Millipore, Burlington, MA, USA). The membranes were incubated with primary antibodies against STAT3, ERK, AKT, p-STAT3, p-ERK, p-AKT, gp130, and p-gp130 (1:1000 dilution; Cell Signaling Technology, Danvers, MA, USA), and β-actin (1:4000 dilution; Cell Signaling Technology) overnight. After incubation, the membranes were treated with horseradish peroxidase-conjugated secondary antibodies (1:4000 dilution; Cell Signaling Technology), and protein bands were detected using Super-Signal West Femto Maximum Sensitivity Substrate (Thermo Fisher Scientific). Chemiluminescent signals were analyzed using a ChemiDoc XRS gel imaging system (BioRad Laboratories, Hercules, CA, USA). Western blot bands from three experiments were quantitated using the ImageJ software.

## Proliferation assay

MH7A cells were seeded in 96-well plates containing 200 μL of RPMI medium and incubated at 37°C in 5% $CO_2$ for 72 h. After 72 h of incubation with hyper IL-6, the cells were treated with LMT-28 (25 μM), kaempferol (12.5 μM), or combination for 24 h. The absorbance was measured using the D-Plus™ CCK cell viability assay kit (DONGIN LS, Gyeonggi-do, South Korea) at 450 nm.

## Scrape-wound migration assays

MH7A cells were seeded in a 6-well plate at $2\times10^5$ cells/well and wounded using sterilized pipette tips. The tips were placed under UV light for 1 h prior to use. After wounding, detached cells were washed using PBS, and the cells were pre-treated with LMT-28 (25 μM), kaempferol (12.5 μM), or combination for 3 days. After incubation, images were captured at 0, 24, and 48 h, and the migration rate of the cells was calculated based on a reference statistical analysis [34] using the following formula: $[(\text{Area}_{t=0h} - \text{Area}_{t=\Delta h})/\text{Area}_{t=0h}] \times 100\%$.

## Invasion assay

MH7A cells were seeded in a BioCoat Matrigel Invasion Chamber (BD Biosciences) with 500 μL of serum-free RPMI medium containing LMT-28 and/or kaempferol. Normal RPMI medium (700 μL) was placed outside of the chamber. After 48 h of incubation, infiltrating cells were stained using a Differential Quick Stain Kit (Electron Microscopy Sciences, Hatfield, PA, USA). The number of invading cells was quantified using ImageJ.

## Statistical analysis

To evaluate the synergistic effect of the LMT-28 and kaempferol combination, we calculated the coefficient of drug interaction (CDI) according to a reference statistical methodology [35]. CDI was calculated using the following formula:

$$\text{CDI} = \text{AB}/(\text{A} \times \text{B})$$

A synergistic effect is implied when the CDI < 1. Results are presented as mean ± standard error of the mean (SEM). Data were compared for statistical significance using Student's t-test or one-way analysis of variance (ANOVA) with Dunnett multiple comparison test. P-values less than 0.05, 0.01, and 0.001 were considered statistically significant.

# Results

## LMT-28 and kaempferol combination ameliorates synovial inflammation in CIA mice model

The CIA model was established by intradermal injection of collagen and an adjuvant into the tails of DBA1/J mice. Subsequently, mice were orally administered LMT-28 and kaempferol every 2 days for 70 days. The effect of the co-administration of LMT-28 and kaempferol on RA development was evaluated using arthritis scoring and histological analysis of CIA mice. As a result, the arthritis score in the LMT-28 and kaempferol combination group was lower than that in the single administration group (Fig 1A). The combination of LMT-28 and kaempferol effectively suppressed toe swelling and bending in mice (Fig 1B). At day 70 after first immunization, mouse paw tissues were dehydrogenated and embedded in paraffin. The paraffin-embedded slide sections were stained with hematoxylin and eosin to analyze the histological features of the joint tissues of CIA mice (Fig 1C). Combined administration inhibited

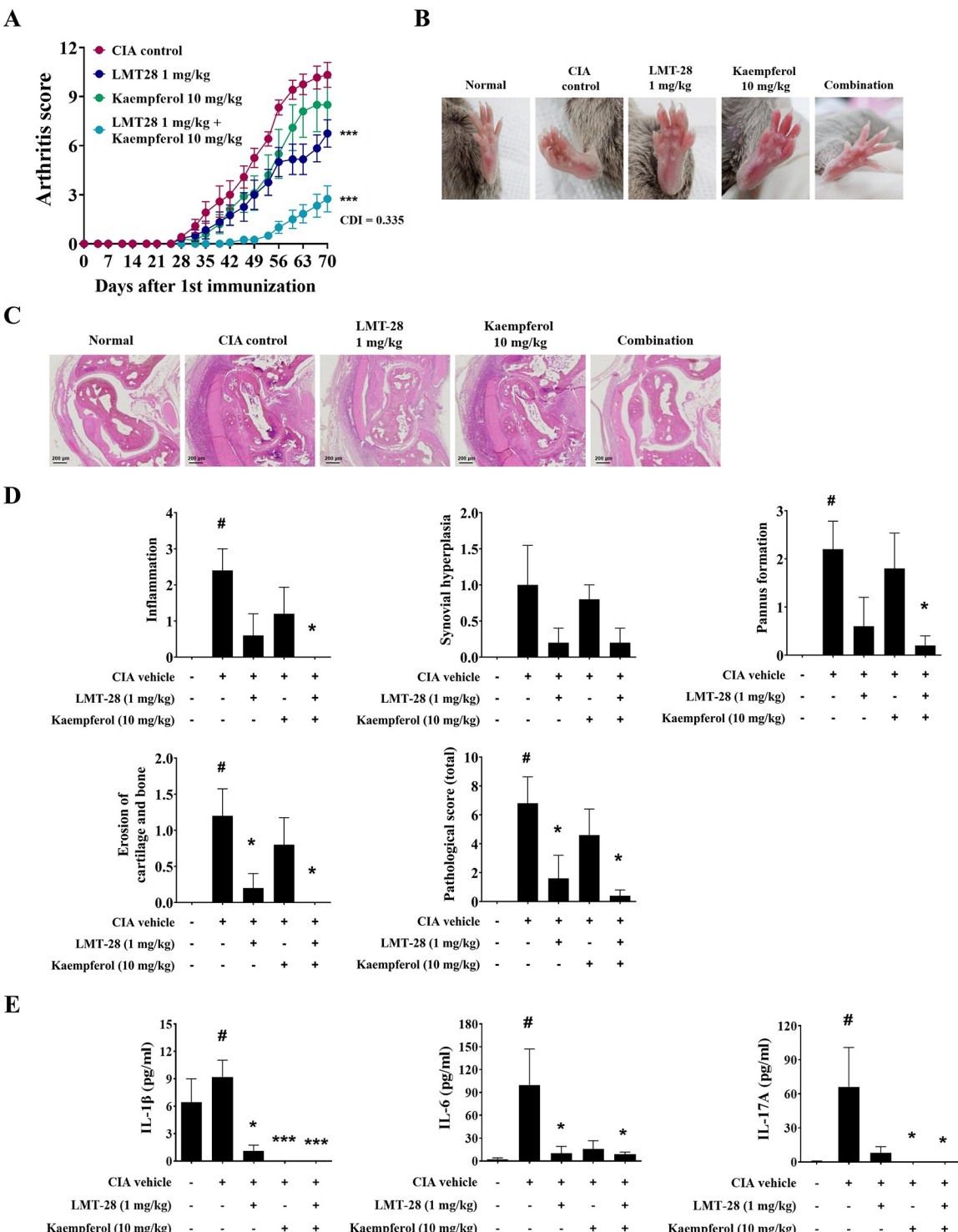

**Fig 1. Co-administration effect of LMT-28 and kaempferol on collagen-induced arthritis mouse model.** (A) Arthritis scores of mice were measured every 3 days starting at 21 days after the first immunization. (B) DBA1/J mice showed deformation and swelling of the toe and ankle joints. (C) Paraffin-embedded sections of CIA mice joint were stained with hematoxylin and eosin. (D) Histological analysis was performed based on the analysis criteria: inflammation, synovial hyperplasia, pannus formation, and erosion of cartilage and bone. (E) The protein levels of IL-6, IL-1β and IL-17 in CIA mouse serum were measured using ELISA. Data are presented as mean ± SEM. #$p < 0.05$ vs. normal group, *$p < 0.05$, **$p < 0.01$, and ***$p < 0.005$ vs. CIA control group.

histopathological phenomena, such as inflammation, synovial hyperplasia, pannus formation, and erosion of cartilage and bone (Fig 1D). Additionally, we analyzed the levels of circulating pro-inflammatory cytokines in the serum of CIA mice. Compared with single treatment, co-administration of LMT-28 and kaempferol down-regulated the serum levels of IL-1β, IL-6 and IL-17A (Fig 1E). Taken together, these results showed that combined administration has a synergistic effect on RA development.

## Combined administration of LMT-28 and kaempferol regulates the T cell differentiation

Based on the anti-inflammatory effects of the LMT-28 and kaempferol combination described in the previous section, we investigated whether LMT-28 and kaempferol has a modulatory effect on T cell populations. Splenocytes isolated from CIA mice were stimulated with PMA and ionomycin, and T cell populations in each group were analyzed using flow cytometry. The population of $CD25^+Foxp3^+$ regulatory T (Treg) cell subsets was upregulated in the spleens of LMT-28 or kaempferol single-treated mice compared with CIA control mice (Fig 2A), but there was no synergistic effect of LMT-28 and kaempferol on Treg differentiation. Additionally, the differentiation of $CD4^+IL-17A^+$ Th17 cells in the spleen increased in the CIA control group compared with the normal group and significantly suppressed in the groups administered LMT-28 alone or in combination with kaempferol (Fig 2B).

To further demonstrate the combined effect of LMT-28 and kaempferol on Th17 cell differentiation, we performed a Th17 cell differentiation assay. The population of Th17 cell differentiated from CD3/CD28-stimulated naïve $CD4^+$ T lymphocytes was downregulated by a single treatment with LMT-28 or kaempferol and was suppressed by co-treatment (Fig 2C). Thus, the co-administration of LMT-28 and kaempferol synergistically inhibited the differentiation of naïve $CD4^+$ T lymphocytes into Th17 cells compared with single treatments.

## Co-administration of LMT-28 and kaempferol inhibits excessive osteoclast differentiation

Excessive differentiation of osteoclasts induces abnormal regeneration of bones and destruction of synovial tissues [36]. In this study, we investigated whether the combined treatment suppresses RANKL-induced osteoclast differentiation. BMMs were stimulated with M-CSF (20 ng/mL) and RANKL (40 ng/mL) and then treated with LMT-28 (25 μM) and/or kaempferol (12.5 μM). TRAP staining result showed that osteoclast activity was downregulated to a greater extent in the LMT-28 and kaempferol combination group than in the single drug treatment groups (Fig 3A). Additionally, the combined treatment suppressed the number of differentiated osteoclasts more effectively than the single drug treatments (Fig 3B). We also analyzed the mRNA expression of the key transcription factors *nfatc1* and *c-fos* and the downstream proteins *ctsk*, and *trap*. Co-administration of LMT-28 and kaempferol suppressed the increase in mRNA expression of osteoclast-related factors in differentiated osteoclasts (Fig 3C). These results suggested that the combination of LMT-28 and kaempferol has a synergistic inhibitory effect on RANKL-induced osteoclastogenesis.

## LMT-28 and kaempferol shows a suppressive effect on the hyperactivation of IL-6-induced signaling pathways in RA-FLS

To analyze the mechanism underlying the combined effect of LMT-28 and kaempferol, we evaluated how the co-treatment affects the IL-6-induced signaling pathway. The human RA-FLS cell line MH7A was pre-treated with LMT-28 (25 μM) and/or kaempferol (12.5 μM)

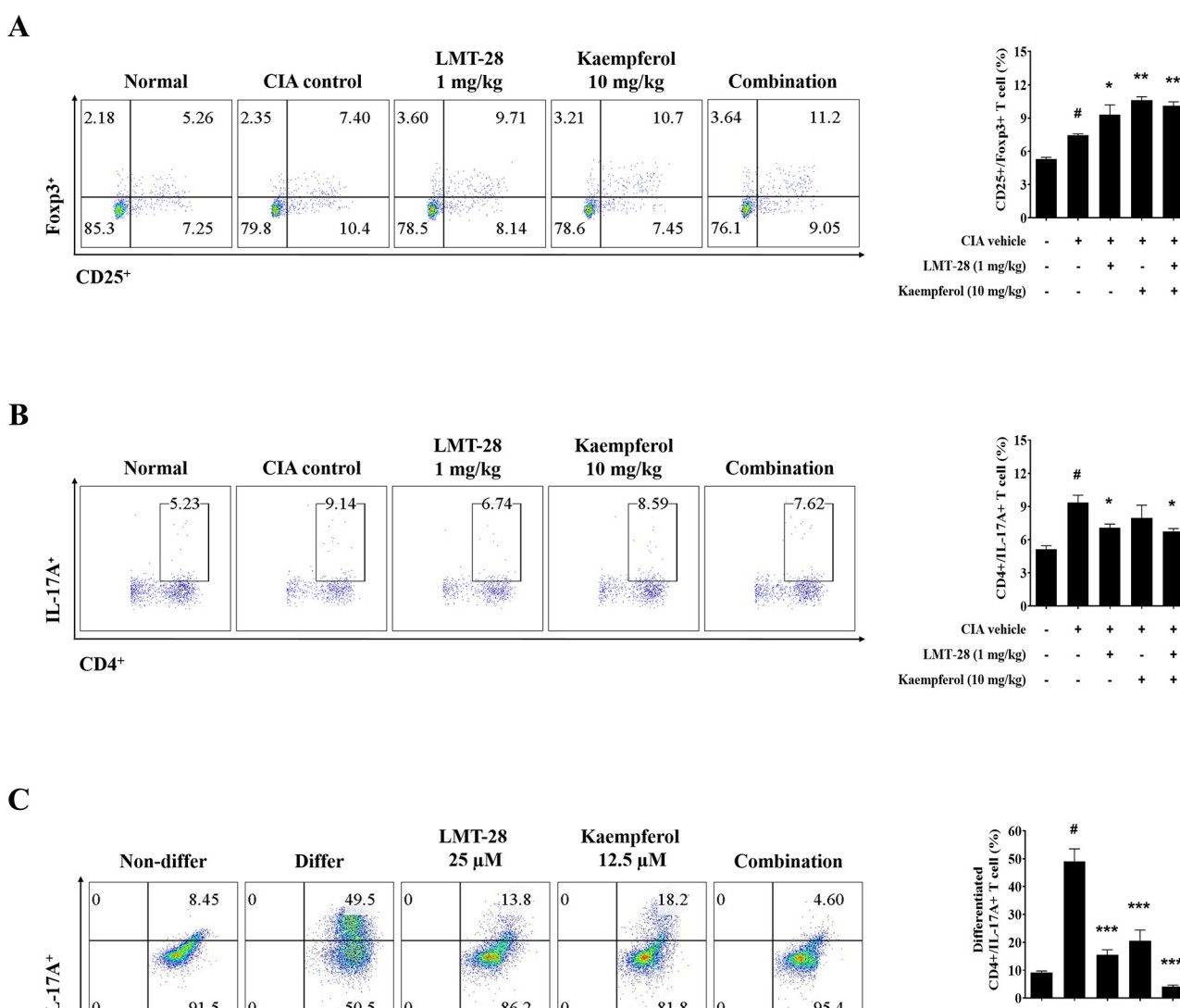

**Fig 2. Effect of LMT-28 and kaempferol on T cell differentiation.** Flow cytometry dot plots (left) and percentage comparisons (right) of (A) Treg cell and (B) Th17 cell population in splenocytes of CIA mice. (C) Dot plots and percentage comparisons of Th17 cell population differentiated from sorted splenocytes. Data are presented as mean ± SEM. #$p<0.05$ vs. normal group or non-treated group, *$p<0.05$, **$p<0.01$, and ***$p<0.005$ vs. control group.

for 1 h, and stimulated with hyper IL-6 (20 ng/mL). Densitometric analysis of gp130, STAT3, ERK, and AKT was performed using western blotting. No significant inhibitory effect on the gp130, STAT3 and ERK signaling pathways in the single treatment groups was observed; however, the combined treatment showed an enhanced inhibitory effect on hyper IL-6-induced phosphorylation of gp130, STAT3, ERK, and AKT compared to single treatment (Fig 4A and 4B). These results suggest that the combination of LMT-28 and kaempferol prevents RA onset and osteoclastogenesis via synergistic inhibition of the IL-6-induced signaling pathway in RA-FLS.

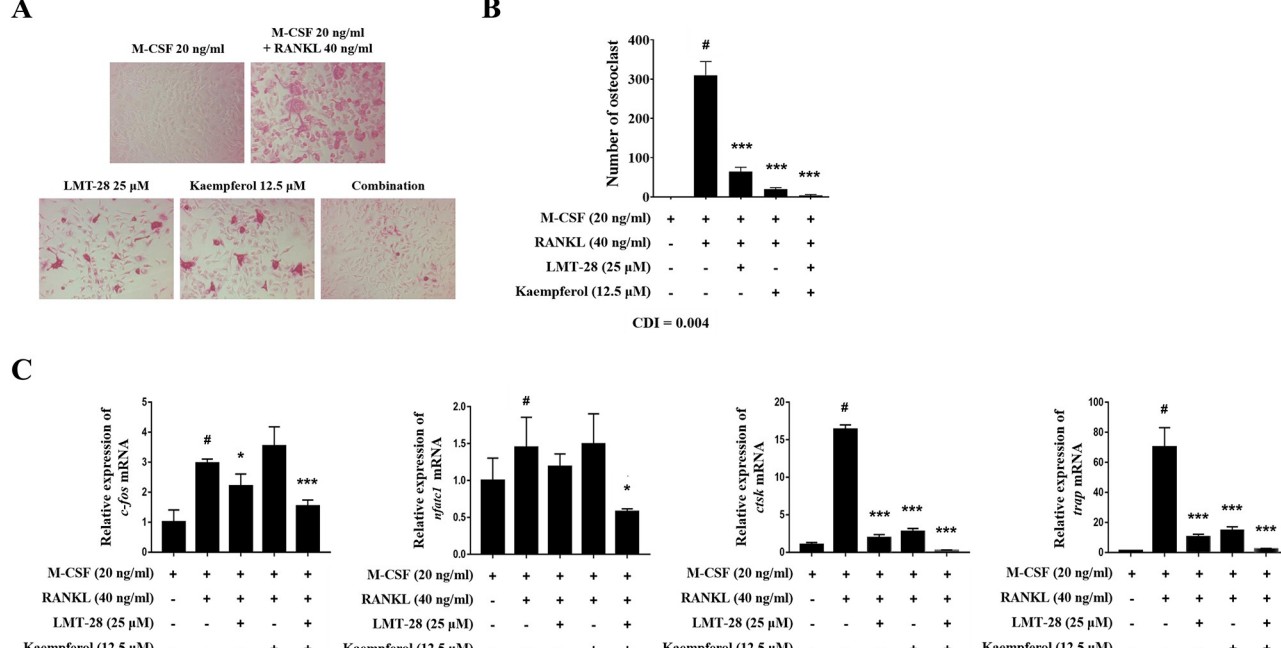

**Fig 3. Combination effect of LMT-28 and kaempferol on RANKL-induced osteoclastogenesis.** BMMs isolated from normal mice were stimulated with M-CSF (20 ng/mL) and/or RANKL (40 ng/mL) and treated with LMT-28 (25 μM) and/or kaempferol (12.5 μM). (A) Microscopic image (200× magnification) of osteoclasts stained for TRAP activity and (B) quantification of the number of TRAP+ multinucleated osteoclasts (≥3 nuclei) in each treatment group are presented. (C) The mRNA levels of *c-fos*, *nfatc1*, *ctsk* and *trap* in osteoclasts were measured. Data are presented as mean ± SEM. #$p<0.05$ vs. M-CSF-treated group; *$p<0.05$, **$p<0.01$, and ***$p<0.005$ vs. M-CSF- and RANKL-treated control group.

## Combination of LMT-28 and kaempferol suppresses RA-FLS hyperactivation and MMP expression

RA-FLS play an important role in synovial inflammation and tissue destruction during RA pathogenesis, and the pro-inflammatory cytokine IL-6 has been demonstrated to induce the hyperactivation of RA-FLS [37]. Based on the enhanced inhibitory efficacy of the co-treatment on the IL-6 signaling pathway activation in RA-FLS, we hypothesized that the combined treatment would have a synergistic negative effect on IL-6-induced hyperactivation of RA-FLS. To validate this theory, MH7A cells were stimulated with hyper IL-6 (20 ng/mL) and subsequently treated with LMT-28 (25 μM) and/or kaempferol (12.5 μM). Compared with single treatment, co-treatment with LMT-28 and kaempferol more effectively inhibited the IL-6-induced RA-FLS proliferation (Fig 5A). Additionally, we investigated the combination effect on RA-FLS migration and invasion. The combined treatment suppressed the IL-6-induced migration rate (Fig 5B) and invasive activity of MH7A cells (Fig 5C).

Furthermore, we measured MMP level to investigate whether LMT-28 and kaempferol could prevent IL-6-induced overexpression of MMPs in RA-FLS. RT-qPCR results showed that the co-treatment of LMT-28 and kaempferol significantly suppressed the mRNA expression of *Mmp-1*, *Mmp-2*, *Mmp-3*, *Mmp-9*, and *Mmp-13* in RA-FLS (Fig 5D). Additionally, the results of ELISA showed that the protein levels of MMP-1, MMP-3, and MMP-13 were downregulated by co-treatment of LMT-28 and kaempferol (Fig 5E). These results show that co-treatment with LMT-28 and kaempferol has an enhanced inhibitory effect on the IL-6-induced hyperactivation of RA-FLS.

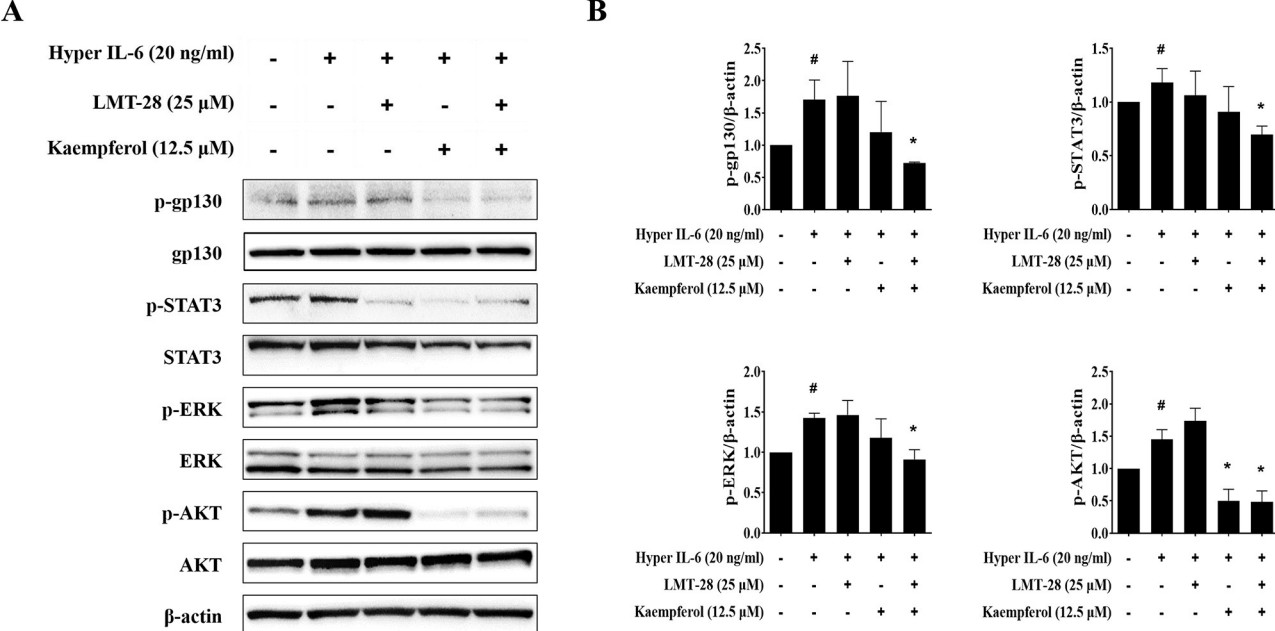

**Fig 4. Effect of LMT-28 and kaempferol combination on IL-6-induced signaling in RA-FLS.** The human RA-FLS cell line MH7A was pre-treated with LMT-28 (25 µM) and/or kaempferol (12.5 µM) for 1 h, and stimulated with hyper IL-6 (20 ng/mL) for 5 min. Total proteins were extracted from MH7A cell homogenates and analyzed using western blotting. (A) Densitometric analysis of western blot bands and (B) expression level graphs of p-gp130, p-STAT3, p-ERK and p-AKT in MH7A cells are presented. Data are presented as mean ± SEM. #$p<0.05$ vs. non-treated group; *$p<0.05$ vs. hyper IL-6 treated group.

## LMT-28 and kaempferol suppresses the hyperactivation of chondrocytes

To further investigate the efficacy of LMT-28 and kaempferol co-treatment on chondrocyte activity, the human chondrocyte cell line C28/I2 was stimulated with hyper IL-6 (20 ng/mL) and then treated with LMT-28 (25 µM) and/or kaempferol (12.5 µM). Hyper IL-6 upregulated the proliferation of chondrocytes, and the combination of LMT-28 and kaempferol suppressed chondrocyte proliferation more effectively than the single treatments (Fig 6A).

Excessive migration and invasion of chondrocytes play crucial role in the degenerative erosion of joints [38]. We examined whether excessive migration and invasion of chondrocytes could be suppressed by co-treatment with LMT-28 and kaempferol. The migration rate of chondrocytes increased in the IL-6-treated group and was effectively suppressed by treatment with LMT-28 and kaempferol (Fig 6B). Additionally, co-treatment neutralized the invasive activity of chondrocytes (Fig 6C). In conclusion, combination of LMT-28 and kaempferol synergistically inhibited chondrocyte hyperactivation.

## Discussion

This study aimed to investigate the synergistic effects of LMT-28 and kaempferol on RA development. In our previous study, a single administration of 5 mg/kg LMT-28 suppressed RA pathology by targeting IL-6 activity in a CIA *in vivo* mouse model. In the present study, we attempted to determine whether a lower dose of LMT-28 co-administered with other drugs has a synergistic effect on suppressing RA development. In this study, we lowered the concentration of LMT-28 to 1 mg/kg and administered it together with kaempferol. Compared with single drug administration, the co-administration of LMT-28 and kaempferol effectively alleviated RA symptoms.

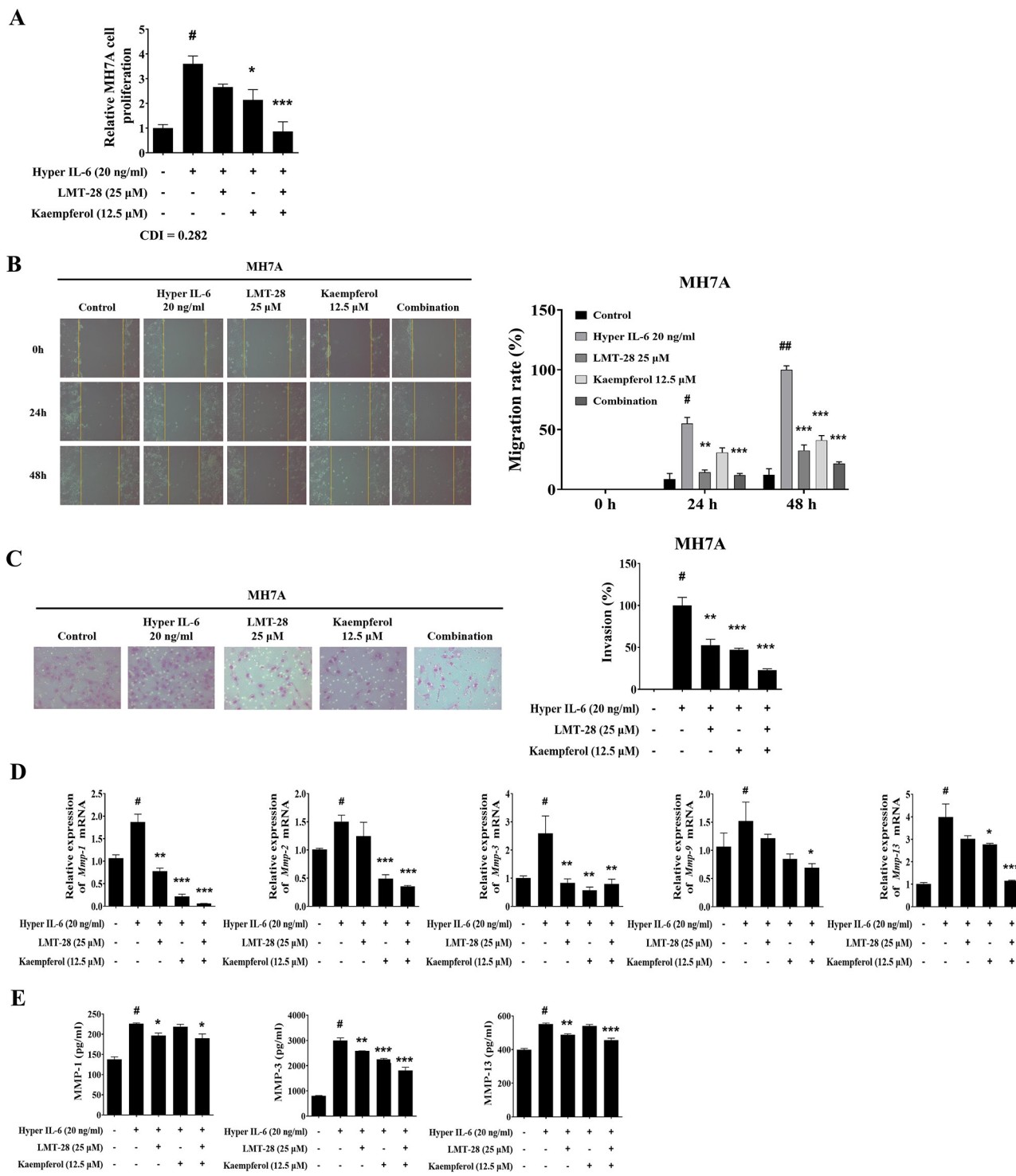

**Fig 5. Effect of LMT-28 and kaempferol co-treatment on RA-FLS activation.** MH7A cells were stimulated with hyper IL-6 (20 ng/mL) and treated with LMT-28 (25 μM) and/or kaempferol (12.5 μM). (A) MH7A cell proliferation was measured using the CCK-8 assay. The migration activity of MH7A cells was evaluated by wound-healing assay. MH7A cells were scratched using a sterile 200 μL pipette tip, followed by treatment with LMT-28 and/or Kaempferol for 48 h. (B, left) Images were captured at 200× magnification, and (B, right) the migration rate of MH7A cells was measured at 0, 24 and 48 h after treatment. Matrigel invasion assay was used to determine invasion after 48 h in LMT-28 and kaempferol-treated MH7A cells. (C, left) Microscopic images (200× magnification) of Matrigel and (C, right) the number of invading cells are presented. Relative (D) mRNA expression and (E) protein levels of MMPs in MH7A cells were measured. Data are presented as mean ± SEM. #$p<0.05$ vs. non-treated control group; *$p<0.05$, **$p<0.01$, and ***$p<0.005$ vs. hyper IL-6 group.

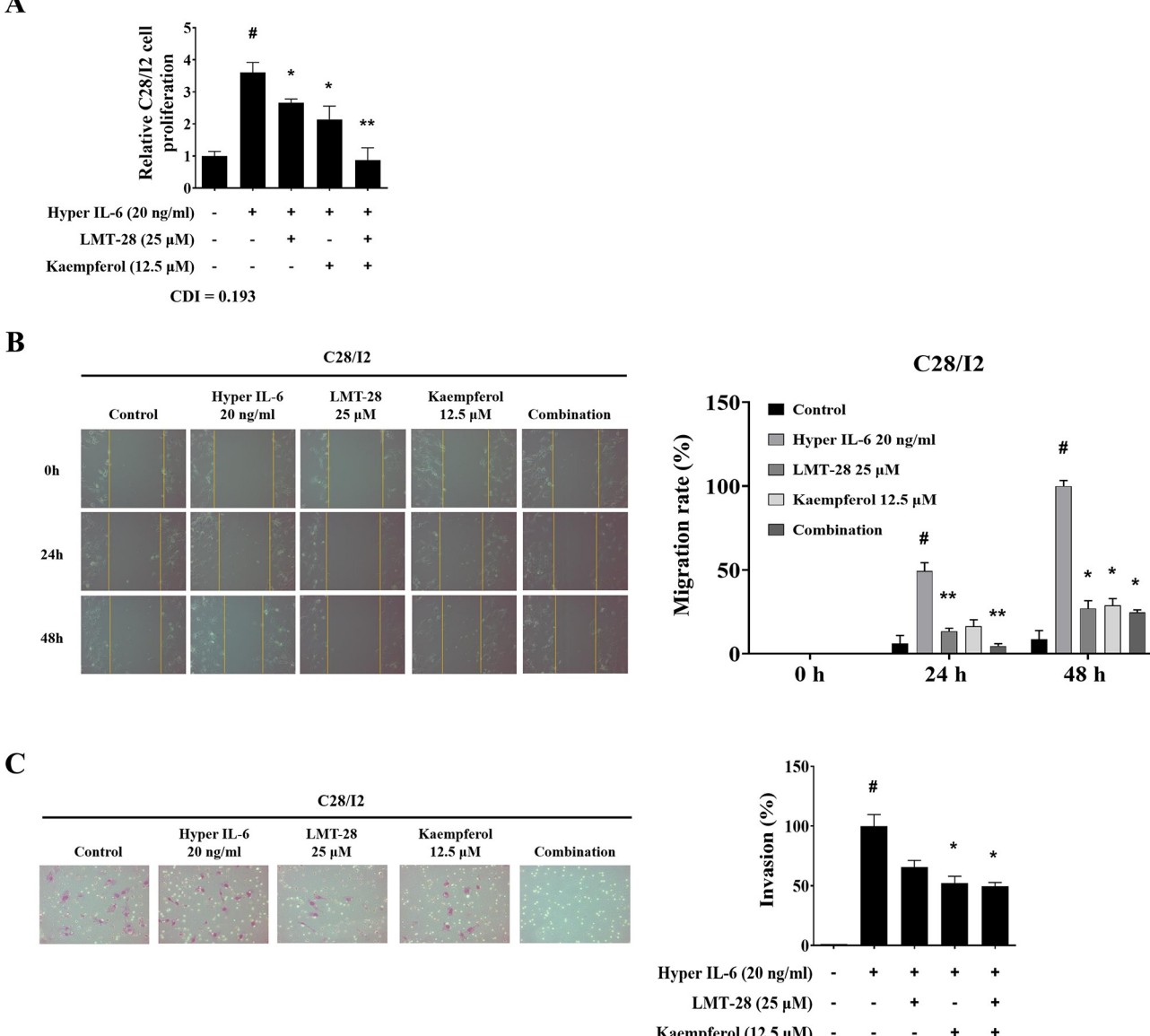

**Fig 6. Combined effect of LMT-28 and kaempferol on chondrocyte activation.** The human chondrocyte cell line C28/I2 was stimulated with hyper IL-6 (20 ng/mL) and treated with LMT-28 (25 µM) and/or kaempferol (12.5 µM). (A) C28/I2 cell proliferation was measured using CCK-8 assay. The migration activity of C28/I2 cells was evaluated by the wound-healing assay. C28/I2 cells were scratched using a sterile 200 µL pipette tip, followed by treatment with LMT-28 and/or kaempferol for 48 h. (B, left) Images were captured at 200× magnification, and (B, right) the migration rate of C28/I2 cells was measured at 0, 24 and 48 h after treatment. Matrigel invasion assay was used to determine invasion after 48 h in LMT-28 and kaempferol treated C28/I2 cells. (C, left) Microscopic images (200× magnification) of Matrigel and (C, right) the number of invading cells are presented. Data are presented as mean ± SEM. #$p < 0.05$ vs. non-treated control group; *$p < 0.05$, **$p < 0.01$, and ***$p < 0.005$ vs. hyper IL-6 group.

T cells are activated by the stimulation of immunomodulatory factors, following which they infiltrate synovial tissues and secrete various inflammatory cytokines during arthritis [39]. This study investigated the combined effect of LMT-28 and kaempferol on T cell differentiation using CIA mice and Th17 cell differentiation using *ex vivo* assay. Our results showed that combined treatment of LMT-28 and kamepferol had an enhanced inhibitory effect on Th17 cell differentiation. Uncontrolled osteolysis of the bone tissue is a typical symptom of arthritis,

and osteoclasts secrete TRAP and CtsK to induce bone resorption, exacerbating RA symptoms [40]. Osteoclast formation is influenced by the network of cytokines, chemokines, and immune cells. Recent studies have used anti-RANKL antibodies such as denosumab to inhibit osteoclast formation [41]. However, anti-RANKL therapies have no anti-inflammatory effects and must be combined with other anti-inflammatory drugs. Therefore, appropriate combined treatment can be an alternative to suppress osteoclastogenesis and RA development. In this study, we demonstrated that the combined effects of LMT-28 and kaempferol on RANKL-induced excessive osteoclast differentiation. The mRNA level of *nfatc1*, a key transcription factor for osteoclast differentiation, was not significantly different in the groups administered LMT-28 and kaempferol alone but was significantly suppressed in the co-treatment group. Additionally, the mRNA expression levels of *c-fos*, *ctsk* and *trap* were down-regulated by the combined treatment. In conclusion, our results showed that the combination of LMT-28 and kaempferol synergistically inhibited excessive osteoclast differentiation and mRNA expression of osteoclast-related genes.

Previous studies have demonstrated that IL-6 overproduction increases RANKL expression in the synovial environment and induces the differentiation of osteoclast precursor cells [42]. Based on the enhanced suppressive effects of the LMT-28 and kaempferol combination on osteoclastogenesis, we hypothesized that this treatment combination may modulate the hyperactivation of IL-6 signaling pathway. To verify our hypothesis, we assessed the combination effect on the hyperactivation of IL-6-induced signaling pathways in the RA-FLS. Single treatment of LMT-28 (25 µM) or kaempferol (12.5 µM) had no significant effect on IL-6-induced signaling pathway activation, but the combined treatment significantly inhibited the phosphorylation of gp130, STAT, ERK, and AKT (Fig 4A and 4B). In a previous study, single treatment of LMT-28 did not effectively inhibit the phosphorylation of signaling molecules induced by IL-6 at concentrations below 50 µM [33], but in this study, the co-administration of 25 µM LMT-28 with kaempferol inhibited the phosphorylation of the signaling molecules. These results demonstrate the improved inhibitory effect of LMT-28 and kaempferol combination on the hyperactivation of IL-6-induced signaling pathways in RA-FLS.

RA-FLS is the main cell type involved in RA pathogenesis, inducing pannus formation and erosion of articular cartilage by invading the synovial tissue [43]. In a previous study, kaempferol inhibited the migration and invasion of RA-FLS by blocking IL-6-induced MAPK signaling pathways [44]. However, the effects of LMT-28 on the migration and invasion of RA-FLS remain unclear. Therefore, we tested the combined effect of LMT-28 and the combination with kaempferol on the IL-6-induced hyperactivation of RA-FLS. In the present study, proliferation, migration, and invasion of RA-FLS were effectively inhibited by LMT-28 treatment and the combination therapy showed an enhanced suppressive effect compared to single treatment (Fig 5A–5C). Since increased expression of MMPs is a hallmark of RA-FLS activation, we measured the mRNA and protein levels of MMPs secreted by RA-FLS. LMT-28 suppressed the mRNA and protein expression of MMP and the combined treatment with kaempferol more effectively downregulated MMP expression (Fig 5D and 5E). These results suggest that the combined administration of LMT-28 and kaempferol may alleviate RA pathology by suppressing the IL-6-induced hyperactivation of RA-FLS.

Furthermore, we investigated whether the combined treatment has an inhibitory effect on chondrocyte activation. Osteoarthritis (OA) is a chronic type of arthritis, whose development is induced by the excessive activation of chondrocytes [45]. Similar to RA-FLS, the migration and invasion activities of chondrocytes induce cartilage damage and loss, resulting in the development of OA [46]. Thus, we assessed the effects of the LMT-28 treatment and co-treatment on chondrocytes activation. As a result, single treatment of LMT-28 down-regulated chondrocyte proliferation, migration, and invasion and the combined treatment of LMT-28

and kaempferol showed an inhibitory enhanced effect (Fig 6A–6C). Considering that LMT-28 and kaempferol inhibit excessive chondrocyte activation, our results suggest that LMT-28 and kaempferol have the potential to suppress OA development. Further studies are needed to evaluate the suppressive effects of LMT-28 on the pathogenesis of OA. Taken together, our findings suggest that combination of LMT-28 and kaempferol synergistically regulates IL-6-induced hyperactivation of synovial cells and can be an alternative option for arthritis treatment.

## Conclusions

Co-administration of LMT-28 and kaempferol inhibited arthritis symptoms by regulating of Th17 cell and osteoclast differentiation. Mechanistic studies on IL-6-induced signaling pathways in RA-FLS suggested that the combination of LMT-28 and kaempferol has a significant effect on hyperactivation of IL-6-related activities in arthritis. Additionally, the combined treatment with LMT-28 and kaempferol had a significant regulatory effect on the hyperactivation of RA-FLS and chondrocytes. The results of our study suggest that the co-administration of LMT-28 and kaempferol is a potential therapy option for arthritis.

## Supporting information

**S1 Raw images. Whole membrane images of western blotting analysis.**
(PDF)

**S1 Raw data.**
(ZIP)

## Author Contributions

**Conceptualization:** Young-Jin Jeong, Yeon-Hwa Park, Hee Jung Kim.

**Data curation:** Young-Jin Jeong, Sun-Ae Park, Yeon-Hwa Park, Lee Kyung Kim.

**Formal analysis:** Young-Jin Jeong, Hae-Ri Lee.

**Funding acquisition:** Hee Jung Kim, Tae-Hwe Heo.

**Investigation:** Young-Jin Jeong, Sun-Ae Park, Yeon-Hwa Park, Lee Kyung Kim.

**Methodology:** Young-Jin Jeong, Sun-Ae Park, Yeon-Hwa Park.

**Project administration:** Hee Jung Kim.

**Supervision:** Tae-Hwe Heo.

**Validation:** Hae-Ri Lee, Hee Jung Kim.

**Visualization:** Young-Jin Jeong, Sun-Ae Park, Yeon-Hwa Park.

**Writing – original draft:** Young-Jin Jeong.

**Writing – review & editing:** Sun-Ae Park, Hae-Ri Lee, Hee Jung Kim.

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
