## [Decision Letter · Decision Letter 0]

7 May 2024

PONE-D-24-12536Anti-inflammatory effect of the combined treatment of LMT-28 and kaempferol in a collagen-induced arthritis mouse modelPLOS ONE

Dear Dr. Heo,

Thank you for submitting your manuscript to PLOS ONE. After careful consideration, we feel that it has merit but does not fully meet PLOS ONE’s publication criteria as it currently stands. Therefore, we invite you to submit a revised version of the manuscript that addresses the points raised during the review process. Please submit your revised manuscript by Jun 21 2024 11:59PM. If you will need more time than this to complete your revisions, please reply to this message or contact the journal office at plosone@plos.org. Please include the following items when submitting your revised manuscript:A rebuttal letter that responds to each point raised by the academic editor and reviewer(s). You should upload this letter as a separate file labeled 'Response to Reviewers'.A marked-up copy of your manuscript that highlights changes made to the original version. You should upload this as a separate file labeled 'Revised Manuscript with Track Changes'.An unmarked version of your revised paper without tracked changes. You should upload this as a separate file labeled 'Manuscript'.If applicable, we recommend that you deposit your laboratory protocols in protocols.io to enhance the reproducibility of your results. Protocols.io assigns your protocol its own identifier (DOI) so that it can be cited independently in the future. For instructions see: https://journals.plos.org/plosone/s/submission-guidelines#loc-laboratory-protocols. Additionally, PLOS ONE offers an option for publishing peer-reviewed Lab Protocol articles, which describe protocols hosted on protocols.io. Read more information on sharing protocols at https://plos.org/protocols?utm_medium=editorial-email&utm_source=authorletters&utm_campaign=protocols.

We look forward to receiving your revised manuscript.

Kind regards,

Masanori A. Murayama

Academic Editor

PLOS ONE

Journal Requirements:

   "This work was supported by the Basic Science Research Program through the National Research Foundation of Korea (NRF) funded by the Ministry of Education, Science, and Technology [grant number. 2018R1A6A1A03025108 and 2022R1A2C2009911]."

7. PLOS ONE now requires that authors provide the original uncropped and unadjusted images underlying all blot or gel results reported in a submission’s figures or Supporting Information files. This policy and the journal’s other requirements for blot/gel reporting and figure preparation are described in detail at https://journals.plos.org/plosone/s/figures#loc-blot-and-gel-reporting-requirements and https://journals.plos.org/plosone/s/figures#loc-preparing-figures-from-image-files. When you submit your revised manuscript, please ensure that your figures adhere fully to these guidelines and provide the original underlying images for all blot or gel data reported in your submission. See the following link for instructions on providing the original image data: https://journals.plos.org/plosone/s/figures#loc-original-images-for-blots-and-gels. 

Additional Editor Comments:

Thank you for submitting your manuscript. It is very interesting. However, I think this manuscript need more discussion. Please revise your manuscript according to my and reviewers comments.

1. In abstract, authors need introduce what is Kaempferol and LMT-28. This manuscript requires detail information about Kaempferol and LMT-28. The therapeutic mechanism including molecular mechanism such as receptor is unknown. At least, authors should discuss about this point.

2. In CIA model, authors should show the histological scores and anti-IIC antibody levels. Please refer Nat Commun. 2021 Jan 4;12(1):94.

3. The inhibitory effect of Kaempferol and LMT-28 in various cells is dose-dependent?

Reviewers' comments:

Reviewer's Responses to Questions

**Comments to the Author**

1. Is the manuscript technically sound, and do the data support the conclusions?

Reviewer #1: Yes

Reviewer #2: Yes

Reviewer #3: Yes

2. Has the statistical analysis been performed appropriately and rigorously? 

Reviewer #1: Yes

Reviewer #2: Yes

Reviewer #3: Yes

3. Have the authors made all data underlying the findings in their manuscript fully available?

Reviewer #1: Yes

Reviewer #2: Yes

Reviewer #3: Yes

4. Is the manuscript presented in an intelligible fashion and written in standard English?

Reviewer #1: Yes

Reviewer #2: Yes

Reviewer #3: Yes

5. Review Comments to the Author

Reviewer #1: PONE-D-24-12536

Anti-inflammatory effect of the combined treatment of LMT-28 2 and kaempferol in a collagen-induced arthritis mouse model

Comments to Authors:

This study presents an intriguing investigation into the combined effects of LMT-28 and kaempferol in addressing rheumatoid arthritis. Given the significant burden autoimmune diseases place on patients, conquering them remains a paramount objective in modern medicine.

The paper is well-written, with clear data presentation and discussion. However, some aspects could benefit from further elaboration:

1) Mechanism of Synergy: The authors should delve deeper into the anti-inflammatory mechanisms of LMT-28 and kaempferol, explaining how their combination leads to superior therapeutic effects compared to individual treatment.

2) Clarifying Figure 1: A clear link needs to be established between Figure 1A's time points and the data presented in Figure 1C.

3) Updated Reference: A recent review on the IL-6/gp130 pathway should be cited to strengthen the argument.

4) Quantifying Histological Findings: In Figure 1C, the analysis of histological features in CIA mouse joint tissue via H&E staining requires a scoring system. The results should be presented graphically alongside the images to enhance clarity.

5) Missing Methodology: The "Materials and Methods" section lacks details regarding the experimental methods used with chondrocytes C28/I2. This needs to be addressed.

Reviewer #2: This study describes the synergistic effects of LMT-28 and kaemppferol on rheumatoid arthritis (RA), particularly on differentiation, migration, and invasion of synovial cells. Overall, the study is well designed and the manuscript is written well. There are several comments:

1. Authors need to discuss more in detail what made you specifically choose kaempferol as a candidate drug to investigate the synergistic effect of LMT-28. How about other anti-inflammatory drugs?

2. In a previous study, LMT-28 was reported to be effective in suppressing the development of RA by inhibiting the activation of the IL-6 signaling pathway in a CIA mouse model. In this manuscript, however, the activation of signaling pathways did not seem to be inhibited by single-treatment of LMT-28. Authors should explain why the effect of LMT-28 on signaling pathway activation in RA-FLS was shown to be different in this study.

3. Authors described that LMT-28 or the combination therapy decreased or improved the incidence of RA. Since the treatment was administered before the onset of RA, it can be considered to be effective in preventing RA. The preventative effect is different from reducing or improving existing arthritis. Please revise these statements throughout the manuscript.

4. Gene nomenclature for all mouse- and human-derived genes, such as transcription factors and MMPs, needs to be modified according to the standard notation.

Reviewer #3: PONE-D-24-12536

Comments to Authors:

This is a well conducted study regarding the combination effect of LMT-28 and kaempferol on collagen-induced arthritis (CIA). The authors showed the suppressive effects of combination treatment with LMT-28 and kaempferol on the progression of arthritis in CIA model. In addition, the combination treatment inhibited Th17-cell differentiation both in vitro and in vivo and activated regulatory T cell proportion in vivo. Combination treatment also inhibited various functions of RA-FLS and chondrocytes in terms of their proliferation, migration, and invasion.

These results collectively indicated that the combination treatment with LMT-28 and kaempferol has a potential therapeutic merit to treat rheumatoid arthritis. In order to make more complete story, authors need to address following points.

1. In the abstract, authors need to choose either combined treatment or combination treatment for consistency.

2. Enhanced effects on signaling pathway / synergistic effects on RA development

-> Enhanced (negative or antagonistic) effects should be make more sense in the context.

3. Line 82, disease activity -> disease severity

4. Line 95, synergistic effects -> potential effects since it is before experiment . Therefore, they do not know whether they have any synergistic effects.

5. Line 98, by regulating -> by downregulating

6. Introduction-Rationale. It is not entirely clear to me why the authors decided to test the LMT-28/kaempferol combination as a potential RA treatment (in vitro/in vivo). Although often different treatments may be combined to produce a greater effect, still the authors need to expand on the underlying biology of the individual compounds and why their combination may yield synergistic effect.

7. Lines 38 – 40 and 324 – 325: the sentences are ambiguous. Can the authors please reformulate these sentences?

8. Lines 334 – 336: the sentences are ambiguous. Can the authors please reformulate these sentences?

9. Lines 301 -302: lack references.

10. In Figure 6, the efficacy of the combination treatment was measured using the chondrocyte C28/I2 cell line. I couldn't find this in the material and methods section. Please add it.

11. Line 324, enhanced effect -> enhanced negative or inhibitory effect

12. Line 336, synergistic effect ->synergistic negative or inhibitory effect

13. Line 362, co-treatment improved the invasive activity of chondrocytes -> co-treatment dampened or neutralized the invasive activity of chondrocytes

14. Line 410, enhanced effect -> enhanced negative or inhibitory effect.

15. It would be better to make all the units as mg/kg instead of mpk.

16. Figure 1A Kaem 10 mpk -> Kaempferol 10 mg/kg

6. PLOS authors have the option to publish the peer review history of their article (what does this mean?). If published, this will include your full peer review and any attached files.

Reviewer #1: No

Reviewer #2: No

Reviewer #3: No

---

## [Author Response · Author response to Decision Letter 0]

28 May 2024

PLoS One

Responses to the reviewer’s comments:

We appreciate the opportunity to revise our manuscript. We had carefully considered all comments and tried to describe in detail and correct all requiring corrections. Our point-wise responses to the reviewers’ comments are given below. We hope that this revision is sufficiently improved the paper will be accepted for publication in PLOS ONE.

Our detailed responses are described below:

Journal Requirements:

1. To comply with PLOS ONE submissions requirements, in your Methods section, please provide additional information regarding the experiments involving animals and ensure you have included details on (1) methods of sacrifice, (2) methods of anesthesia and/or analgesia, and (3) efforts to alleviate suffering.

Response: In the Materials and Methods section, we have added more details about how animals were euthanized and efforts to alleviate suffering in experimental animals.

"This work was supported by the Basic Science Research Program through the National Research Foundation of Korea (NRF) funded by the Ministry of Education, Science, and Technology [grant number. 2018R1A6A1A03025108 and 2022R1A2C2009911]."

Response: We added the following sentence to the cover letter: “The funders had no role in study design, data collection and analysis, decision to publish, or preparation of the manuscript.”.

Response: The retracted references were removed and replaced, detailed below.

1) The existing reference ‘Amarasekara DS, Yun H, Kim S, Lee N, Kim H, Rho J. Regulation of osteoclast differentiation by cytokine networks. Immune network. 2018;18(1).’ was replaced with ‘Amarasekara DS, Kim S, Rho J. Regulation of osteoblast differentiation by cytokine networks. International journal of molecular sciences. 2021;22(6):2851.’ and numbered [11].

2) The retracted reference ‘Martel-Pelletier J, Barr AJ, Cicuttini FM, Conaghan PG, Cooper C, Goldring MB, et al. Osteoarthritis. Nature reviews Disease primers. 2016;2(1):1-18.’ was replaced with ‘Goldring MB. The role of the chondrocyte in osteoarthritis. Arthritis & Rheumatism: Official Journal of the American College of Rheumatology. 2000;43(9):1916-26.’ and numbered [45].

 

Additional Editor Comments:

1. In abstract, authors need introduce what is Kaempferol and LMT-28. This manuscript requires detail information about Kaempferol and LMT-28. The therapeutic mechanism including molecular mechanism such as receptor is unknown. At least, authors should discuss about this point.

Response: Thank you for your comment. We added the detailed information about LMT-28 and kaempferol in abstract section.

2. In CIA model, authors should show the histological scores and anti-IIC antibody levels. Please refer Nat Commun. 2021 Jan 4;12(1):94.

Response: Thank you for your comment. We conducted histological analysis in the CIA model and attached the revised data to the manuscript (Fig 1D). 

In a previous study, it was confirmed that the increased level of anti-CII IgG in CIA control mouse serum was significantly reduced in the mouse group administered LMT-28. The results of the study were published in the Journal of Immunology in 2015 (Hong, Soon-Sun, et al. "A novel small-molecule inhibitor targeting the IL-6 receptor β subunit, glycoprotein 130." The Journal of Immunology (2015) 195(1): 237-245). Based on the effect of LMT-28 administration confirmed in previous studies, it is expected that anti-CII IgG production will be inhibited by co-administration of LMT-28 and kaempferol in CIA model. The combination of LMT-28 and kaempferol effectively inhibited toe swelling and bending in mice and inhibited histopathological phenomena, such as inflammation, synovial hyperplasia, pannus formation, and erosion of cartilage and bone in CIA mouse model. In addition, the serum levels of IL-1β, IL-6 and IL-17A in the serum of CIA mice down-regulated by combination of LMT-28 and kaempferol. These results show that the combination of LMT-28 and kaempferol synergistically inhibits synovial inflammation in the CIA mouse model.

3. The inhibitory effect of Kaempferol and LMT-28 in various cells is dose-dependent?

Response: In a previous study, LMT-28 dose-dependently inhibited cell proliferation of IL-6 dependent TF-1 cell (Hong, Soon-Sun, et al. "A novel small-molecule inhibitor targeting the IL-6 receptor β subunit, glycoprotein 130." The Journal of Immunology 2015, 195(1): 237-245). Additionally, LMT-28 inhibited hyper IL-6-induced cell proliferation of MH7A cell (Park, Yeon‐Hwa, Hee Jung Kim, and Tae‐Hwe Heo. "A directly GP130‐targeting small molecule ameliorates collagen‐induced arthritis (CIA) by inhibiting IL‐6/GP130 signalling and Th17 differentiation." Clinical and Experimental Pharmacology and Physiology 2020, 47(4): 628-639).

Kaempferol also showed dose-dependent inhibitory effects on various cell types. In a previous study, kaempferol dose-dependently inhibited IL-6-induced COX-2 expression and STAT3 phosphorylation in THP-1 cell (Basu, Anandita, et al. "STAT3 and NF-κB are common targets for kaempferol-mediated attenuation of COX-2 expression in IL-6-induced macrophages and carrageenan-induced mouse paw edema." Biochemistry and biophysics reports 2017, 12: 54-61). Additionally, kaempferol suppressed the MAPK pathway activation and CIA symptoms in a dose-dependent manner (Pan, Dongmei, et al. "Kaempferol inhibits the migration and invasion of rheumatoid arthritis fibroblast-like synoviocytes by blocking activation of the MAPK pathway." International immunopharmacology 2018, 55: 174-182).

We previously used MH7A cells to test whether kaempferol could suppress the proliferation of RA-FLS by preliminary experiments. Kaempferol significantly suppressed the proliferation of MH7A cells in a dose-dependent manner. Co-treatment with LMT-28 and kaempferol in in a dose-dependent manner effectively inhibited the IL-6-induced MH7A cell proliferation (under figure).

  

Review Comments to the Author

Reviewer #1

1. Mechanism of Synergy: The authors should delve deeper into the anti-inflammatory mechanisms of LMT-28 and kaempferol, explaining how their combination leads to superior therapeutic effects compared to individual treatment.

Response: Thank you for comments. In the Introduction section, we added the following explanation as to why we designed the combined treatment of LMT-28 and kaempferol in this study. 

“Kaempferol, used as a combination partner of LMT-28 in this study, is a type of flavonoid and is reported to have anti-inflammatory and anticancer effects in various immune diseases. Additionally, in a previous study, kaempferol inhibited the migration and invasion activity of RA-FLS by inhibiting the activation of the MAPK pathway. Based on previous research, this study designed the combined administration of LMT-28 and kaempferol to confirm further improved anti-arthritis effects by simultaneously targeting the JAK/STAT pathway and MAPK pathway, which are major signaling pathways that play an important role in arthritis development.”.

2. Clarifying Figure 1: A clear link needs to be established between Figure 1A's time points and the data presented in Figure 1C.

Response: The histological analysis described in Fig 1C were performed at the day 70 after the first immunization. As you suggested, we added a description to the results section of Figure 1C.

3. Updated Reference: A recent review on the IL-6/gp130 pathway should be cited to strengthen the argument.

Response: Based on your comment, we cited the most recent review paper related to the IL-6/gp130 pathway (Ding, Qian, et al. "Signaling pathways in rheumatoid arthritis: implications for targeted therapy." Signal transduction and targeted therapy 8.1 (2023): 68.) and reference numbered [24].

4. Quantifying Histological Findings: In Figure 1C, the analysis of histological features in CIA mouse joint tissue via H&E staining requires a scoring system. The results should be presented graphically alongside the images to enhance clarity.

Response: We agree with your opinion. We conducted histological analysis in the CIA model and attached the revised data to the manuscript (Fig 1D). The combination of LMT-28 and kaempferol effectively inhibited histopathological phenomena, such as inflammation, synovial hyperplasia, pannus formation, and erosion of cartilage and bone in CIA mouse model. Table 2 was added including histopathological scoring criteria: Inflammation, synovial hyperplasia, pannus formation, erosion of cartilage and bone.

5. Missing Methodology: The "Materials and Methods" section lacks details regarding the experimental methods used with chondrocytes C28/I2. This needs to be addressed.

Response: We added information about the C28/I2 cell line to the Materials and Methods section.

 

Reviewer #2

1. Authors need to discuss more in detail what made you specifically choose kaempferol as a candidate drug to investigate the synergistic effect of LMT-28. How about other anti-inflammatory drugs?

Response: Thank you for your comment. We added the reasons for designing the combination of LMT-28 and kaempferol in the Introduction section. In our previous study, we investigated whether the combined treatment of LMT-28 and metformin has synergistic inhibitory effect on arthritis in CIA mouse model (Park, Yeon-Hwa, et al. "Combination of LMT-28 and metformin improves beneficial anti-inflammatory effect in collagen-induced arthritis." Pharmacology 2021, 106(1-2): 53-59). Additionally, combined treatment of TNF-targeting tetrahydropapaverine (THP) and LMT-28 suppressed RA development via inhibiting Th17 cell and osteoclast differentiation (Park, Yeon-Hwa, et al. "Combination of gp130-targeting and TNF-targeting small molecules in alleviating arthritis through the down-regulation of Th17 differentiation and osteoclastogenesis." Biochemical and biophysical research communications 2020, 522(4): 1030-1036). There was synergistic effect with LMT-28 in combination with other anti-inflammatory agents, metformin and THP in CIA model. However, previous studies did not confirm inhibitory effects on cellular activities such as RA-FLS and chondrocyte migration and invasion. Therefore, in this study, we aimed to confirm the synergistic inhibitory effect on synovial cell hyperactivation through combined treatment with kaempferol.

2. In a previous study, LMT-28 was reported to be effective in suppressing the development of RA by inhibiting the activation of the IL-6 signaling pathway in a CIA mouse model. In this manuscript, however, the activation of signaling pathways did not seem to be inhibited by single-treatment of LMT-28. Authors should explain why the effect of LMT-28 on signaling pathway activation in RA-FLS was shown to be different in this study.

Response: Thank you for your comment. We added the comment in the discussion section. Detailed below:

“In a previous study, single treatment of LMT-28 did not effectively inhibit the phosphorylation of signaling molecules induced by IL-6 at concentrations below 50 μM, but in this study, the co-administration of 25 μM LMT-28 with kaempferol inhibited the phosphorylation of the signaling molecules. These results demonstrate the improved inhibitory effect of LMT-28 and kaempferol combination on the hyperactivation of IL-6-induced signaling pathways in RA-FLS.”

3. Authors described that LMT-28 or the combination therapy decreased or improved the incidence of RA. Since the treatment was administered before the onset of RA, it can be considered to be effective in preventing RA. The preventative effect is different from reducing or improving existing arthritis. Please revise these statements throughout the manuscript.

Response: This study is a prevention model in which LMT-28 and kaempferol were administered before the onset of arthritis. Applying your comments, we appropriately revised the descriptions such as ‘reduce’ and ‘decrease’.

4. Gene nomenclature for all mouse- and human-derived genes, such as transcription factors and MMPs, needs to be modified according to the standard notation.

Response: We have modified the gene nomenclature for the mouse and human -derived genes in the revised manuscript in accordance with the standard notation.  

Reviewer #3 

1. In the abstract, authors need to choose either combined treatment or combination treatment for consistency.

Response: Thank you for your comment. For consistency of expression, the sentences were modified to ‘combined treatment’.

2. Enhanced effects on signaling pathway / synergistic effects on RA development

-> Enhanced (negative or antagonistic) effects should be make more sense in the context.

Response: Thank you for your comment. Applying your comments, descriptions in this manuscript were revised to make more sense.

3. Line 82, disease activity -> disease severity

Response: The 'Disease Activity' description was modified to 'Disease Severity'.

4. Line 95, synergistic effects -> potential effects since it is before experiment. Therefore, they do not know whether they have any synergistic effects.

Response: The expression 'synergistic' was removed and the description was modified.

5. Line 98, by regulating -> by downregulating

Response: The word has been modified.

6. Introduction-Rationale. It is not entirely clear to me why the authors decided to test the LMT-28/kaempferol combination as a potential RA treatment (in vitro/in vivo). Although often different treatments may be combined to produce a greater effect, still the authors need to expand on the underlying biology of the individual compounds and why their combination may yield synergistic effect.

Response: Thank you for your comment. In the Introduction section, we described the biological mechanism by which the combination of LMT-28 and kaempferol exerts synergistic effects. Detailed below:

“Kaempferol, used as a combination partner of LMT-28 in this study, is a type of flavonoid and is reported to have anti-inflammatory and anticancer effects in various immune diseases. Additionally, kaempferol inhibited the migration and invasion activity of RA-FLS by inhibiting the activation of the MAPK pathway. In our previous study, we demonstrated that LMT-28 exerts a therapeutic effect on RA development by inhibiting the IL-6-induced signaling pathways; however, it remains unclear whether LMT-28 has a regulatory effect on RA-FLS migration and invasion. Based on previous research, this study designed the combined administration of LMT-28 and kaempferol to confirm further improved preventive effects of combination on the hyperactivation of RA-FLS and RA development by simultaneously targeting the JAK/STAT pathway and MAPK pathway, which are major signaling pathways that play an important role in arthritis development.”.

7. Lines 38 – 40 and 324 – 325: the sentences are ambiguous. Can the authors please reformulate these sentences?

Response: We revised the sentences.

8. Lines 334 – 336: the sentences are ambiguous. Can the authors please reformulate these sentences?

Response: We revised the sentences.

9. Lines 301 -302: lack references.

Response: We added the reference (Ref #36. Boyle, William J., W. Scott Simonet, and David L. Lacey. "Osteoclast differentiation and activation." Nature 2

---

## [Decision Letter · Decision Letter 1]

2 Jun 2024

Anti-inflammatory effect of the combined treatment of LMT-28 and kaempferol in a collagen-induced arthritis mouse model

PONE-D-24-12536R1

Dear Dr. Heo,

We’re pleased to inform you that your manuscript has been judged scientifically suitable for publication and will be formally accepted for publication once it meets all outstanding technical requirements.

Kind regards,

Masanori A. Murayama

Academic Editor

PLOS ONE

Additional Editor Comments (optional):

Thank you for re-submitting your manuscript. I am grad for sending you this letter. Your manuscript is remarkable for publish. Congratulations.

Reviewers' comments:

Reviewer's Responses to Questions

**Comments to the Author**

1. If the authors have adequately addressed your comments raised in a previous round of review and you feel that this manuscript is now acceptable for publication, you may indicate that here to bypass the “Comments to the Author” section, enter your conflict of interest statement in the “Confidential to Editor” section, and submit your "Accept" recommendation.

Reviewer #1: All comments have been addressed

Reviewer #2: All comments have been addressed

2. Is the manuscript technically sound, and do the data support the conclusions?

Reviewer #1: Yes

Reviewer #2: Yes

3. Has the statistical analysis been performed appropriately and rigorously? 

Reviewer #1: Yes

Reviewer #2: Yes

4. Have the authors made all data underlying the findings in their manuscript fully available?

Reviewer #1: Yes

Reviewer #2: Yes

5. Is the manuscript presented in an intelligible fashion and written in standard English?

Reviewer #1: Yes

Reviewer #2: Yes

6. Review Comments to the Author

Reviewer #1: The author's revisions and responses to major revisions have all proven to be scientifically valid and reliable.

Reviewer #2: Authors properly made responses to reviewer's comments and revised the manuscript well, which is most likely acceptable for publication.

7. PLOS authors have the option to publish the peer review history of their article (what does this mean?). If published, this will include your full peer review and any attached files.

Reviewer #1: No

Reviewer #2: No

---

## [Editor Report · Acceptance letter]

22 Jul 2024

PONE-D-24-12536R1 

PLOS ONE

Dear Dr. Heo, 

I'm pleased to inform you that your manuscript has been deemed suitable for publication in PLOS ONE. Congratulations! Your manuscript is now being handed over to our production team.

Kind regards, 

on behalf of

Dr. Masanori A. Murayama 

Academic Editor

PLOS ONE